# Remote Sensing of Ocean Fronts in Marine Ecology and Fisheries

**Igor M. Belkin**

College of Marine Science and Technology, Zhejiang Ocean University, Zhoushan 316022, China;
igormbelkin@gmail.com

**Abstract:** This paper provides a concise review of the remote sensing of ocean fronts in marine ecology and fisheries, with a particular focus on the most popular front detection algorithms and techniques, including those proposed by Canny, Cayula and Cornillon, Miller, Shimada et al., Belkin and O'Reilly, and Nieto et al.. A case is made for a feature-based approach that emphasizes fronts as major structural and circulation features of the ocean realm that play key roles in various aspects of marine ecology.

**Keywords:** ocean front; marine ecology; fisheries; front detection; satellite imagery; feature-based approach

## 1. Introduction

The seemingly continuous, smoothly varying ocean is full of discrete features such as fronts and eddies. Living creatures seek out such oceanic features as habitats of choice for foraging, reproduction, recruitment, and migration. By providing habitats for a variety of species during different ontogenetic stages and seasons, ocean fronts and eddies play key roles in the ecology of marine animals. However, the bulk of oceanographic data used in marine ecology and fisheries consist of oceanic variables (including observable and derived properties) measured or computed at given points of four-dimensional ocean space (latitude, longitude, depth, and time) and not explicitly linked to any objects or elements of the ocean's structure. Discrete ocean features such as fronts and eddies are always inexplicitly present in such data sets; however, information about such features (their location, structure, and environmental variables) is typically not extracted from data, and therefore not explicitly used. The feature-based approach to oceanographic data management consists of two major steps: (1) feature detection in oceanographic data; (2) feature data usage in research and applications. This survey focuses on (a) front detection in satellite data and (b) applications of front data in marine ecology and fisheries.

Remote sensing from Earth observation satellites confers tremendous benefits to fisheries. The high flight altitude (usually above 500 km) of such satellites allows instant coverage of huge expanses of the ocean; the satellite orbital speed of ~8 km·s$^{-1}$ combined with the sun-synchronous orbits of some satellites ensures frequent global coverage (e.g., four times daily for NOAA satellites) with repeat observations of a given point every 12 h at the same local time; the high spatial resolution of modern sensors allows observations of sub-mesoscale (<10 km), small-scale (<1 km), and micro-scale (<100 m) details of the ocean's structure and circulation; and finally the concurrent usage of multiple sensors that measure different ecologically important characteristics has a synergetic potential to be exploited by researchers and practitioners.

The above-listed benefits conferred by satellites were realized by fishermen and fisheries managers long ago. Maps of sea surface temperature (SST) based on satellite measurements became widely available in the 1960s and 1970s, being routinely generated by space agencies, meteorological centers, and fisheries agencies in several countries, e.g., USA, UK, Canada, Japan, Australia, France, and Russia. Initially, such maps were

broadcast by radio and available as analogue facsimiles of maps produced manually by experts. With the advent of the Internet, such maps are now generated automatically and objectively by computers and are broadcast in digital form.

At the same time, even the early rudimentary satellite-based SST maps documented the most salient ocean circulation features, such as fronts and eddies. Visual observations from satellites and orbital manned stations have corroborated and reinforced such findings. The large- and mesoscale circulation features such as fronts and eddies have been traditionally recognized as ecologically important by marine biologists, while fishermen have long exploited their potential as fishing grounds, albeit largely in coastal ocean areas. Now that satellites afford a global view of the ocean, this information is available for deep-sea fisheries as well.

Despite the commonly acknowledged ecological importance of major circulation features, particularly fronts and eddies, their use in everyday activities and long-term planning of marine fisheries remained limited until recently. The main impediment to the wide practical acceptance of the feature-based approach was the lack of computer software for efficient and effective processing of vast amounts of satellite data, including computer algorithms for feature extraction from satellite imagery. This situation has changed quite dramatically over the last 10 years.

This review focuses on remote sensing of fronts and applications of front detection in marine ecology and fisheries oceanography. According to a commonly accepted definition [1,2], a front is a relatively narrow zone of enhanced horizontal gradients of water properties (temperature, salinity, nutrients, etc.) that separates broader, relatively uniform areas with different water masses or different vertical structures (stratification). Fronts occur on a variety of space–time scales, from a few meters up to thousands of kilometers horizontally and up to kilometers vertically, and from days to millions of years, respectively. Most fronts are seasonally persistent, with some fronts being prominent in all seasons. Cross-frontal ranges of temperature and salinity can be as large as 15 °C and 3 ppt, while typical ranges are 2 to 5 °C and 0.3 to 1.0 ppt, respectively [1,2]. Fronts are the main structural elements of the oceanic realm, impacting all trophic levels across a wide range of space–time scales [2–17]. Fronts are loci of enhanced primary and secondary production and higher concentrations of plankton aggregated by convergence toward fronts. Dense aggregations of phytoplankton and zooplankton attract planktivorous schooling fish, squid, and large piscivorous fish. Life stages of many fish species are associated with fronts, including spawning, nursing, foraging, and migrations.

The sheer and ever-growing amount of satellite data necessitated the development of objective automated methods of front detection in satellite imagery. In the 1990s, a global survey of ocean fronts from satellite data (supported by NASA) began at the Graduate School of Oceanography of the University of Rhode Island (GSO/URI), based on a sophisticated state-of-the-art front detection algorithm developed by Jean-François Cayula and Peter Cornillon [18–20]. The Cayula–Cornillon algorithm (CCA) was used to generate a unique global front data base, allowing an unprecedented view of World Ocean fronts at sub-mesoscale resolution in space (presently down to 1 km) and synoptic resolution in time (nominally every 12 h), and their evolution at seasonal, interannual, and decadal scales from 1982 to date [2]. A long-term collaboration between the URI and NOAA since 2005 has resulted in the development of a novel front detection algorithm by Igor Belkin and John O'Reilly [21], named BOA hereafter, which was adopted by NOAA in 2013.

Both algorithms developed at the URI (histogram-based CCA and gradient-based BOA) are now widely used in ecological studies and marine fisheries. Meanwhile, a few alternative algorithms and techniques have been proposed and used for front detection and characterization in oceanography, marine ecology, and fisheries. As a result, researchers and practitioners have a choice of front detection algorithms. It is time to review the history of applications of front detection algorithms in marine ecology and fishery research. At the same time, this review makes a case for the feature-based approach to marine ecology and

fishery research, since fronts are the most ecologically important discrete physical features of the ocean realm. The structure of this paper is as follows:

1.　Section 2 covers satellite missions and sensors, summing up the most important Earth observation satellite missions and sensors relevant to marine ecology and fisheries;
2.　Section 3 covers remote sensing in marine fisheries, providing a retrospective of remote sensing applications in marine fisheries and ecology;
3.　Section 4 covers the ecological role of fronts and eddies, providing a concise review of ecological studies of links between marine animals and major oceanic circulation and structural features, namely fronts and eddies;
4.　Section 5 covers front detection in satellite imagery, presenting five front detection, visualization, and characterization algorithms and methods that have a history of successful applications in marine ecology and fisheries research;
5.　Section 6 covers a feature-based approach to remote sensing in marine ecology and fisheries, which is the central part of this review, as it makes a case for feature detection and application of feature-linked remote sensing data;
6.　Section 7 covers the Discussion, which points out a few scientific and technical problems, suggests a few solutions, draws preliminary conclusions, and outlines perspectives;
7.　Section 8 provides the Summary, which sums up the major takeaway messages.

## 2. Satellite Missions and Sensors

Sea surface temperatures (SSTs) are measured with space-borne infrared (IR) and microwave (MW) radiometers. The advent of the satellite era, marked by the launch of the first artificial Earth satellite, the Russian Sputnik-1 in 1957, was followed from 1967 by launches of weather satellites with IR radiometers that measured SST, a key ecological variable. Systematic high-quality measurements of SST began in 1981 (and have continued uninterrupted to date) with the Advanced Very High-Resolution Radiometers (AVHRR) flown on NOAA polar-orbiting satellites [22]. Currently, the spatial resolution of modern space-borne radiometers (e.g., the Visible Infrared Imaging Radiometer Suite—VIIRS) is better than 1 km, while temporal resolution (12 h) is dictated by their sun-synchronous orbits. High-quality AVHRR SST data provided by the Pathfinder project are available from 1982 to date (https://www.ncei.noaa.gov/products/avhrr-pathfinder-sst; accessed on 18 February 2021). Geostationary satellites' payload sensors acquire images of the ocean as often as every five minutes.

Sea surface chlorophyll (CHL) concentrations are estimated from satellite ocean color data. Such data became widely available after the launch of the Coastal Zone Color Scanner (CZCS) aboard the Nimbus-7 satellite (1978–1986), followed by the Sea-Viewing Wide Field-of-View Sensor (SeaWiFS) aboard the OrbView-2 (aka SeaStar) satellite (1997–2010). The Moderate-Resolution Imaging Spectroradiometers (MODIS) have provided uninterrupted measurements of SST and CHL from the Terra satellite since 1999 and from the Aqua satellite since 2002. The Visible Infrared Imaging Radiometer Suite (VIIRS) that measures SST and CHL among other observables was launched aboard the Suomi NPP satellite in 2011 and NOAA-20 satellite in 2017. The CHL data provided by the above instruments resolve CHL fronts associated with collocated SST fronts, and often (especially in summer) reveal fronts that all but disappear in SST imagery because of the lack of thermal contrast across such fronts.

Sea surface height (SSH) has been systematically continuously measured since 1992 in multiple satellite altimeter missions, including TOPEX/Poseidon, Jason, ERS, Envisat, Cryosat, Sentinel, and SARAL missions. The SSH data provided by modern satellite altimeters resolve major large-scale fronts and many mesoscale fronts.

Sea surface salinity (SSS) has been measured by the Soil Moisture and Ocean Salinity (SMOS) satellite from 2009 until present, the Aquarius radiometer aboard the SAC-D satellite in 2011–2015, and the Soil Moisture Active Passive (SMAP) satellite since 2015. The

satellite SSS data resolve strong salinity fronts in the open ocean and river plume fronts in the coastal ocean.

Sea surface roughness (SSR) is measured by space-borne synthetic aperture radars (SAR) flown in numerous satellite missions (Radarsat, ERS, Envisat, ALOS, TerraSAR-X, etc.). All SARs have a ground resolution of less than 100 m, typically a few tens of meters. Some SARs have a resolution of up to 1 m (TerraSAR-X). Ocean fronts are often visible at the sea surface as lines of ripples. Such lines manifest in SAR imagery as bright lines due to the elevated SSR of the ripples [23]; however, such lines are only visible in light winds.

Visible-spectrum high-resolution imagery of the coastal ocean has been provided by the Landsat mission since 1972. The entire global archive of Landsat imagery is now freely available (Landsat Homepage, Landsat Science (nasa.gov; accessed on 18 February 2021)). The spatial resolution of Landsat imagery is 15 to 60 m, while the temporal resolution is 16 days.

## 3. Remote Sensing in Marine Fisheries

Satellite SST and CHL data that became readily available in 1982 and 1997, respectively, immediately found applications in marine fisheries. Because of the paramount importance of water temperature to marine biota, the first three decades of satellite oceanography (1960s to 1980s) saw an explosion of interest in satellite SST mapping and applications in fisheries [22,24–34]. The satellite imagery from the Coastal Zone Color Scanner (CZCS) flown on Nimbus-7 were used by fisheries oceanographers early on [25]. With the advent of systematic continuous satellite observations of ocean color on a global scale, ocean color data have been used to estimate CHL concentration and primary productivity; these results have been promptly used in marine fisheries [30–32]. Numerous modeling studies of fish stocks have used statistical models in which SST and CHL featured as key ecological variables.

The past applications of SST and CHL data in marine fisheries were largely driven by (1) the data's free availability and (2) the ecological importance of water temperature and chlorophyll concentration. Research-quality SST data from AVHRR flown on NOAA polar orbiting satellites have been available since 1982 owing to the NOAA Pathfinder project. These data allow complete global coverage every 12 h. SST maps based on these data were made freely available from NASA and NOAA at progressively better resolutions of 9, 4, and 1 km. Cloud contamination remains a major problem, since infra-red radiometers cannot see through clouds. By compositing SST images over a week or two, most clouds can be eliminated (except cloud decks that persist for months); however, most important features (fronts and eddies) become blurred.

Satellite maps of CHL, once available, were quickly adopted for use in marine fishery research [24,27,29–36]. A continuous uninterrupted global archive of CHL maps has been available from NASA (https://oceancolor.gsfc.nasa.gov/; accessed on 18 February 2021) since the launch of SeaWiFS in 1997. CHL data have been widely used since the launch of MODIS on Terra (1997) and Aqua (2002) satellites. These data are freely available from NASA (https://oceancolor.gsfc.nasa.gov/; accessed on 18 February 2021) and NOAA (https://www.nodc.noaa.gov/SatelliteData/OceanColor/; accessed on 18 February 2021) at the same spatial and temporal resolutions as the SST data described above. Cloud contamination hampers CHL imagery in the same way as it corrupts SST imagery. The intercalibration of CHL data provided by different sensors and satellite missions remained a problem, which was addressed and largely mitigated under the Ocean Colour Climate Change Initiative project [37].

Satellite SSH data have been applied in marine fisheries and ecology since the 1990s [38–40]. The main benefit of SSH vs. SST and CHL is that SSH is an integral of the vertical density structure of a water column (or vertical integral). Therefore, SSH maps reveal dynamic features such as eddies and fronts better than SST and CHL maps. Another important advantage of SSH data is their all-weather nature, because radar altimeters can see through clouds.

The availability of satellite SST and CHL data, including the continuous uninterrupted global archives of SST (since 1982) and CHL (since 1997), combined with the inherent simplicity of their use as scalar gridded data sets, facilitated the wide acceptance of SST and CHL as environmental variables in marine ecology and fisheries research. Meanwhile, remote sensing applications have progressed from mapping of SST and CHL toward feature detection and usage. Two classes of discrete ocean features stand out owing to their ecological importance and ubiquity, namely fronts and eddies.

## 4. Ecological Roles of Fronts and Eddies

The World Ocean consists of water masses separated by fronts that structure the seemingly continuous, smoothly varying ocean into discrete habitats. In this respect, oceanic fronts are functionally similar to aquatic–terrestrial or purely terrestrial transition zones, also called fronts or ecotones.

Fronts are the most ecologically important circulation and structural features of the ocean realm [3–17,41–48]. Each large-scale front is an ecotone defined as a narrow transition zone between two adjacent marine ecosystems. Each of these ecosystems inhabits a distinct water mass characterized by a unique combination of temperature, salinity, nutrients, and microelements. Most large-scale fronts are strongest at the surface. They extend several hundred meters deep, thereby affecting distributions of oceanic variables in surface and subsurface layers. Some large-scale fronts are strongest in subsurface or intermediate layers. Finally, there are examples of deep or abyssal fronts that affect benthic ecosystems. Owing to their vertical extent to the order of 1 km, most large-scale fronts separate different water mass assemblies that feature distinct vertical structure or stratification types.

Fronts affect various aspects of the ecology of marine animals—their migrations, foraging, reproduction, recruitment, and connectivity [9,11,12,15,42,43,46,49–61]. Fronts are hot spots of marine life, as they often are loci of maximum biodiversity and elevated primary and secondary production [44,46,51,52,62]. The abundance of most fish peaks at or near fronts [3,14,63–68]. For example, using a fluid dynamics approach to ecosystem modeling of the anchovy–sardine regimes and salmon abundances in the California Current, Woodson and Litvin [14] showed that fronts positively affect total ecosystem biomass, production, and fishery abundance and yield.

Most large-scale and mesoscale fronts are known to feature enhanced primary and secondary productivity and elevated biodiversity. The biomass increase observed in fronts is often caused by current convergence, which concentrates passive plankters such as phytoplankton and small zooplankton. Even in the absence of convergence, phytoplankton can grow in fronts [69].

Water mass fronts in oligotrophic seas sometime feature elevated primary productivity (manifested in high CHL) owing to upwelling of nutrient-rich subsurface waters in frontal zones. This phenomenon was observed in the South China Sea (SCS hereafter), where intrusions of oligotrophic Kuroshio waters spurred CHL growth, as reported by Guo et al. [70] (p. 11,565): "Strong fronts due to Kuroshio intrusion and interactions with the SCS water are associated with intense upwelling, supplying high nutrients from the subsurface SCS water and increasing phytoplankton productivity in the frontal region."

The elevated concentration of phytoplankton and small zooplankton attracts large zooplankton and small fish, the latter being preyed upon by larger piscivorous fish and apex predators. However, as noted by Woodson and Litvin [14] (p. 1710), "the effects of fronts such as fishery productivity and biogeochemical cycling hotspots have not been included in models that assess fisheries production . . . "

Another ecologically important aspect of fronts is their role as biodiversity hotspots defined as regions with elevated biodiversity [46,52,62]. Biodiversity typically peaks in fronts, although exceptions are reported when biodiversity at a front is lower than biodiversity in two adjacent water masses separated by the front. For example, of 46 fish taxa identified in the Patagonian Shelf Large Marine Ecosystem by Alemany et al. [71],

"demersal fish diversity increased at the tidal front of Península Valdés but decreased in the frontal zones of the Southern Shelf-Break and Magellan frontal systems" [71] (p. 2111).

The affinity of fish to ocean fronts has long been known to fishermen. In particular, the close association between fish and coastal fronts has been exploited by artisanal fishermen from time immemorial. The affinity of herring stocks to fronts in the Nordic Seas between Iceland and Norway was probably exploited by Norsemen since medieval times. There is little doubt that European fishermen discovered strong fronts of the North Atlantic Current and Labrador Current—they had to cross these currents and fronts—during their regular fishing expeditions to the Grand Banks of Newfoundland, where they fished for Atlantic cod since the early 16th century [72]. In Japan, Michitaka Uda has published numerous papers on fronts and fisheries, including his PhD thesis published as a journal paper in English [73], in which he cited numerous descriptions of ocean fronts known in Japan as "siome" or "shiome", which means a "junction line between two sea currents, a line where two ocean currents meet" (https://jlearn.net/Dictionary/Browse/1953800-shiome; accessed on 18 February 2021).

A strong link between fronts and distribution and abundance of pelagic fishes has been reported in many studies worldwide and has been exploited by pelagic fisheries for a long time. A similar link between fronts and demersal fishes received less attention. Meanwhile, in some regions, e.g., in the Argentine Sea, it is the demersal fishes that are most important economically and ecologically, while local fronts are found to be preferable fishing areas for demersal fisheries, as shown by the distribution of fishing fleets and fishing effort [74]. The strong link between demersal fisheries and fronts of the Argentine Sea is paralleled by the strong concentration of scallop fisheries along the Shelf-Break Front that borders the Patagonian Shelf [75,76]. The prominent role of the Shelf-Break Front in the ecology of scallops is not surprising because this front is anchored at depth by bathymetry (shelf break). The stability of the Shelf-Break Front's near-bottom part (as opposed to the high variability of the surface manifestation of this front) ensures the stability of the benthic habitat, which is apparently beneficial to the scallops.

The above-mentioned ample historical evidence notwithstanding, the link between fronts and fish has not been firmly established until recently. Two factors have contributed to this recent change. The first factor is the current abundance and free availability of satellite data on SST, CHL, and SSH mentioned in the previous section. The second factor is the proliferation of computer algorithms for front detection mentioned in the next section.

Occasionally, especially in earlier studies (but also in some most recent studies), an elevated SST gradient was used as a front proxy [77,78]; thus, fronts (high-gradient zones) were effectively, if not explicitly, considered. However, systematic use of front data in marine fisheries has not occurred until recently, despite the commonly acknowledged ecological importance of fronts. Fortunately, the situation is changing quite rapidly.

The spatial and temporal affinity of fish to fronts is species-specific and is strongly dependent on ontogenetic stage. For example, in a study of Patagonian hoki *Macruronus magellanicus* on the southern Patagonian shelf, Alemany et al. [66] found that younger fish (juveniles) preferred fronts where they preyed upon zooplankton, whereas larger fish (adults) showed no affinity to fronts, as they preyed on larger items. Alemany et al. [66] (p. 191) concluded: "The positive relationship observed between small-sized fish and fronts may be related to the low trophic level of these fish and the abundance of small-sized prey in frontal zones."

Mesoscale eddies are the second most important type of discrete dynamic physical features (after fronts) that affect the ecology of marine animals. Definitions of mesoscale eddies vary [79,80]. It is commonly acknowledged that mesoscale eddies are the largest moving whirls encountered in the ocean (besides a few quasi-stationary gyres), with typical ranges of their parameters as follows: diameter, 50–200 km; vertical extent, 500–1000 m; orbital speed, 50–100 cm·s$^{-1}$; propagation speed, 3–10 km·day$^{-1}$; and lifetime, 1 month–2 years. Ecological roles played by the mesoscale eddies are in some respect similar to the ecological roles played by fronts. Indeed, such eddies often feature sharp gradients along

their periphery. These peripheral zones around mesoscale eddies satisfy all three major criteria of a front, as they are (1) high-gradient zones that (2) divide different water masses and (3) separate different types of vertical structures (stratification) inside and outside of the eddies.

As pointed out by Braun et al. [81] (p. 17187): "Debate regarding how and why predators use fronts and eddies, for example as a migratory cue, enhanced forage opportunities, or preferred thermal habitat, has been ongoing since the 1950s. The influence of eddies on the behavior of large pelagic fishes, however, remains largely unexplored". The combination of remote sensing of eddies with satellite tracking of animal movements has the potential to elucidate the impact of eddies on the ecology of fishes, as exemplified by a recent study of blue sharks in the Gulf Stream region [81]. It turned out that the sharks seek out anticyclonic (AC) eddies, particularly the interiors of AC eddies, for foraging. A satellite tracking study of juvenile loggerhead turtles in the Southwest Atlantic by Gaube et al. [82] revealed a similar tendency, as the turtles tend to congregate in the interiors of AC eddies, apparently to forage, despite low CHL concentrations in surface layers of these eddies. These observations (and some previous [83] and more recent [84] studies as well) show that AC eddies are not lifeless ocean deserts, as commonly thought in the past. Conversely, mesoscale AC eddies support the most abundant mesopelagic fish community in the World Ocean [81]. The above findings dovetailed with a remote sensing study of loggerhead turtles in the Azores region by Chambault et al. [85] that revealed the turtles' affinity to the inner cores of old AC eddies, supposedly due to the higher productivity of old, decaying AC eddies. The above results [81,82,85] that point to the higher productivity of the interiors of AC eddies contradict the observations reported by Godø et al. [83], who found a fish biomass minimum in the centers of three AC eddies in the Norwegian Sea. In the Mozambique Channel, Tew Kai and Marsac [86] found good foraging conditions for frigatebirds and tuna along edges of mesoscale cyclonic eddies and suggested that these conditions arise from the aging process of the cyclonic eddies and their interactions with other eddies. Analyzing fisheries data from the NE Indian Ocean, Arur et al. [87] found that higher catches were associated with the periphery of anticyclonic and the cores of cyclonic eddies (supposedly due to upwelling), while lower catches were associated with the periphery of cyclonic and the cores of anticyclonic eddies (supposedly due to downwelling).

The contradictory results cited above warrant further investigations of the fine-scale horizontal and vertical distributions of productivity in mesoscale eddies.

Occasionally, CHL fronts develop at SST fronts around eddies. In the South China Sea, Ye et al. [88] documented SST fronts at peripheries of mesoscale anticyclonic eddies. These peripheral SST fronts had a stronger influence on surface CHL than seasonal coastal and permanent offshore SST fronts. The CHL enhancement along peripheral SST fronts around eddies is ascribed to "wind pump" upwelling generated by a typhoon that passed over the eddies [88]. In the NE Pacific off British Columbia, a combined analysis of several years of SST and CHL imagery allowed the author to identify a large well-defined CHL ring collocated with a warm SST ring (AC eddy); the CHL ring has likely developed over the pre-existing warm AC ring.

Even though ecological roles played by fronts and eddies are often quite similar, as noted above, automated detection of fronts and eddies from satellite data is typically performed with radically different algorithms. Following the publication of the seminal paper by Chelton et al. [79], automated eddy detection from satellite data has grown into a mature field, with a multitude of eddy detection and tracking algorithms that are often based on different principles [80]. Thus, eddy detection from satellite data warrants a separate review. In this paper, we focus solely on front detection and its applications in marine ecology and fisheries.

## 5. Front Detection in Satellite Imagery

The potential and importance of satellite observations of thermal and color (CHL) fronts for marine ecology and fisheries were recognized early on [6,17,28,30–32,35,36,42,43,74]. Frontal maps are now generated by computer algorithms that detect fronts in satellite images of oceanic variables. Since fronts are narrow, high-gradient zones in satellite images, the task of front detection is equivalent to edge detection in image processing. This terminology is consistent with ocean fronts being boundaries that separate different water masses; thus, each front is a water mass edge. In general-purpose image processing, the tasks of edge detection and image segmentation (into uniform segments) are often complementary. In oceanography, the task of front detection in satellite imagery of oceanic variables can often be considered as complementary to the task of water mass identification, especially regarding large-scale fronts such as the Gulf Stream or Kuroshio.

Various approaches to front detection (or edge detection) in oceanography have been suggested and implemented (for a short review, see the Introduction in [21]). The brief survey below covers algorithms and techniques that were used multiple times in marine ecology and fisheries research; as such, these approaches are de facto established. The most popular front detection algorithms and techniques (listed chronologically) are those proposed by Canny [89], Cayula and Cornillon [18–20], Shimada et al. [90], Belkin and O'Reilly [21], Miller [91], and Nieto et al. [92]. Not all of them are independent: Miller [91] and Nieto et al. [92] are based on Cayula and Cornillon [19].

**Canny algorithm:** The gradient approach is based on the most common and widely accepted definition of fronts as high-gradient zones separating relatively uniform water masses. The main disadvantage of the gradient approach is the noisiness of the gradient field, since differentiation effectively amplifies any noise present in the data. Various algorithms have addressed this disadvantage. The most popular all-purpose gradient-based algorithm for edge detection in 2D imagery was developed by Canny [89]. The Canny algorithm is implemented in Matlab, IDL, Python, C++, R, and Java. It is widely used to detect fronts in SST imagery [93–101]. The Canny algorithm is also used to detect fronts in CHL imagery [102,103] and in SAR imagery, where some fronts manifest as bright lines of elevated sea surface roughness [104,105].

**Cayula–Cornillon algorithm (CCA):** The most sophisticated edge detection algorithm in satellite oceanography was developed by Jean-François Cayula and Peter Cornillon first for SST imagery [18–20] and later successfully applied to CHL imagery. The Cayula–Cornillon algorithm (CCA) is based on a histogram approach; a histogram of SST values of all pixels within an image of a front separating two water masses M1 and M2 would always have two modes corresponding to the water masses M1 and M2, while the front is a locus of SST values that correspond to a minimum between the two modes. At the URI, the CCA has been applied to all available AVHRR SST imagery since 1982, thereby creating a unique global archive of frontal maps that have allowed a global survey of SST fronts to be conducted [2,106].

The CCA is available as a single-image edge detector (SIED) [19] and multi-image edge detector [20]. The CCA gained wide popularity and has been used in numerous studies to map SST fronts globally [2,106] and locally in the following regions:

**Atlantic Ocean:** Baltic Sea [107], Mid-Atlantic Bight [108–110], Georges Bank [111], Gulf of Maine [112], Western North Atlantic [63], Western Iberian Basin [113–115], Sargasso Sea [116], Canadian coastal waters [117], West Florida Shelf [102], North Atlantic Current region [118].

**Pacific Ocean:** Marginal and coastal seas [119], Bering Sea [120–122], Okhotsk Sea [123], Gulf of Alaska [120], California Current [124,125], East China Seas [126], South China Sea [127,128], Tasman Sea [129], Chilean Northern Patagonia [130].

**Indian Ocean:** Red Sea [131], North Indian Ocean [48,103,132,133].

**Arctic Ocean:** Chukchi Sea [120], Beaufort Sea [120,134], Canadian coastal waters [117].

**CCA applications to CHL imagery:** Even though the CCA was originally developed to detect fronts in SST imagery, it has been successfully applied to CHL imagery as well [110,124,135].

**CCA availability:** Since 2010, the CCA-SIED has become publicly available as an R code included in the Marine Geospatial Ecology Tools (MGET) developed by Jason Roberts and collaborators at the Marine Geospatial Ecology Laboratory, Duke University [136]. Owing to the free access to CCA-SIED as part of MGET (available at http://mgel.env.duke.edu/tools; accessed on 18 February 2021), the CCA gained even more popularity and has been used in numerous marine ecological studies reviewed in the next section.

**Miller analysis:** Miller and collaborators developed a novel and powerful technique based on combining and compositing frontal maps of different oceanic variables, e.g., SST and CHL [91,137–139]. Miller's composite frontal map analysis has been used in the marine ecological studies reviewed in the next section.

**Nieto algorithm:** The original CCA-SIED algorithm was modified by Nieto, Demarcq, and McClatchie and applied to the Canary Current System [92]. The Nieto algorithm uses a combination of multiple sliding windows and significantly improves the CCA-SIED performance. The Nieto algorithm has been used in a few studies, including marine ecological studies [140–143].

**Shimada algorithm:** Vázquez et al. [144] applied an entropic approach to edge detection in SST images. This approach (first proposed by Barranco et al. [145]) uses the Jensen–Shannon divergence [146] as a criterion of separation between histograms generated by a window sliding over an image. The Vázquez et al. algorithm [144] was modified and applied in satellite oceanography by Shimada and collaborators [90]. The Shimada et al. algorithm [90] has been used in a few studies of fronts in the China Seas, including the marine ecology and fisheries studies reviewed in the next section.

**Belkin and O'Reilly algorithm (BOA):** Belkin (URI) and O'Reilly (NOAA) developed a novel gradient-based algorithm and applied it to SST and CHL [21]. The main novelty of the Belkin–O'Reilly algorithm (BOA) is a shape-preserving, scale-sensitive, contextual median filter applied selectively and iteratively until convergence. This filter eliminates noise while preserving stepwise fronts and two features endemic to CHL, namely (a) roof edges corresponding to CHL enhancement at hydrographic fronts and (b) peaks corresponding to point-wise CHL blooms. Both features are common in CHL imagery. The BOA is universal, as it can be applied to any scalar oceanic variable. For instance, in a study of the NE Pacific fronts off the British Columbia, the BOA was successfully applied to SST, CHL, SSH, and satellite radar imagery (Gary Borstad, ASL Borstad Remote Sensing Inc., Canada, personal communication). The BOA was originally developed in Matlab and converted to IDL by Jay O'Reilly [21]. The BOA IDL code has been converted to C++ and officially adopted by NOAA in 2013. Frontal CHL maps generated by the BOA C++ code are freely available from NOAA as an operational ocean color product (https://www.ncei.noaa.gov/access/metadata/landing-page/bin/iso?id=gov.noaa.nodc:CoastWatch-OC-Frontal; accessed on 18 February 2021), accessible via the Comprehensive Large-Array Data Stewardship System (CLASS) (https://www.avl.class.noaa.gov/saa/products/search?sub_id=0&datatype_family=OC_FRONTAL&submit.x=25&submit.y=2; accessed on 18 February 2021).

**BOA R code by Galuardi:** The BOA pseudocode published by Belkin and O'Reilly [21] stimulated the implementation of BOA in various computer languages. Ben Galuardi published the BOA pseudocode in R and made his BOA-R code publicly available (https://github.com/galuardi/boaR; accessed on 18 February 2021).

**BOA applications:** The BOA was released to numerous groups around the world after its publication in 2009 and has been successfully used to map ocean fronts in various regions, including the East China Sea [147], Yellow Sea [148,149], South China Sea [70,150], Kuroshio Current [151], and British Columbia waters. The BOA applications in ecological and fisheries studies are reviewed in the next section.

## 6. Feature-Based Approach to Remote Sensing in Marine Ecology and Fisheries

The ever-growing recognition of the paramount importance of circulation features such as fronts and eddies combined with the development and validation of various algorithms for feature detection and tracking from satellite data have profoundly transformed this entire field and led to the incorporation of fronts and eddies into dynamic ocean management [14,16,17,74,129,152,153].

Table 1 sums up remote sensing studies of the ecological roles of fronts, conducted with the most popular front detection and mapping algorithms and techniques (chronologically: Canny [89], Cayula and Cornillon [18–20], Miller [91,137], Shimada et al. [90], Belkin and O'Reilly [21], and Nieto et al. [92]). Also included in Table 1 are remote sensing, marine ecology, and fisheries studies that used a simple gradient calculation in lieu of the more sophisticated front detection or mapping techniques listed above. The inclusive approach to Table 1 is justified because the studies based on a simple gradient calculation could be extended and upgraded by using one of several advanced front detectors that are now freely available.

**Table 1.** Remote sensing of fronts for marine ecology and fisheries.

| Reference (First Author, Ref. No.) | Region | Front Detection Algorithm/Technique | Species |
|---|---|---|---|
| Abdullah [154] | Arabian Sea | CCA-SIED | Tuna |
| Austin [155] | NE Atlantic off UK | CCA-SIED-MGET; Miller | Basking shark |
| Bedriñana-Romano [130] | Chilean Patagonia | CCA-SIED-MGET | Blue whale |
| Bigelow [78] | North Pacific | Gradient | Swordfish, blue shark |
| Bogazzi [75] | Patagonian Shelf | Gradient | Patagonian scallops |
| Braun [59] | North Atlantic | BOA-RG | Sharks |
| Brigolin [156] | Alboran Sea | CCA-SIED; Miller | Various fishes |
| Brodie [157] | Tasman Sea | CCA | Dolphinfish, kingfish |
| Byrne [40] | North Atlantic | BOA-RG | Shortfin mako shark |
| Camacho [158] | California Current | CCA-SIED | Whales and dolphins |
| Chakraborty [103] | North Indian Ocean | CCA-SIED; Canny | Various species |
| Chambault [57] | Gulf Stream | Gradient | Leatherback turtle |
| Chambault [58] | Baffin Bay, Hudson Bay | Gradient | Bowhead whale |
| Chen X.J. [159] | NW Pacific | Gradient | Neon flying squid |
| Cox [56] | Celtic Sea | CCA-SIED; Miller | Gannets |
| Dalla Rosa [64] | NE Pacific | CCA-SIED-MGET | Humpback whale |
| Dell [160] | Tasman Sea | CCA-SIED | Yellowfin tuna |
| Dodge [161] | Northwest Atlantic | BOA-RG | Leatherback turtle |
| Druon [51] | Mediterranean Sea | Gradient | Atlantic bluefin tuna |
| Druon [162] | Arctic Ocean | Gradient | Various species |
| Ebango Ngando [163] | Mauritanian upwelling | CCA-SIED | Mackerel, sardinella |
| Etnoyer [49] | NE Subtropical Pacific | Gradient | Blue whales, turtles |
| Francis [164] | Indian Ocean | CCA-SIED; Canny | Potential fishing zone |
| Friedland [165] | US Northeast Shelf | Gradient | Various species |
| Glembocki [166] | Patagonian Shelf | Gradient | Patagonian red shrimp |
| Haberlin [167] | Celtic Sea | CCA-SIED-MGET | Gelatinous zooplankton |
| Herron [77] | Gulf of Mexico | Gradient | Butterfish |



**Table 1.** *Cont.*

| Reference (First Author, Ref. No.) | Region | Front Detection Algorithm/Technique | Species |
|---|---|---|---|
| Hidayat [168] | Bone Gulf, Indonesia | CCA-SIED | Skipjack tuna |
| Hsu [169] | West & Central Pacific | BOA | Skipjack tuna |
| Hua [170] | NW Pacific | Gradient | Pacific saury |
| Jakubas [60] | Svalbard | CCA-SIED | Little auks |
| Jishad [171] | Bay of Bengal | Gradient | Various species |
| Kulik [172] | Northwest Pacific | BOA-RG | Pacific saury |
| Lan [173] | South Indian Ocean | Shimada | Albacore tuna |
| Lennert-Cody [174] | Eastern Tropical Pacific | CCA-SIED | Bigeye tuna |
| Liao [39] | East China Sea | Shimada | Swordtip squid |
| Liu [175] | East China Seas | CCA-SIED-MGET | Anchovy |
| Louzao [176] | Western Mediterranean | CCA-SIED-MGET | Cory's shearwater |
| Luo [55] | Western North Atlantic | BOA | Tuna, marlin, sailfish |
| Mauna [76] | Patagonian Shelf | Gradient | Patagonian scallop |
| Mazur [177] | US Northeast Shelf | CCA | American lobster |
| Miller [178] | NE Atlantic | CCA-SIED; Miller | Basking shark |
| Mitchell [179] | English Channel | CCA-SIED | Blue shark |
| Mugo [180] | NW Pacific | CCA-SIED-MGET | Tuna, squid, saury |
| Nieblas [140] | SE Tropical Indian Ocean | CCA-SIED-Nieto; Canny | Southern bluefin tuna |
| Nieto [142] | NE Pacific | CCA-SIED-Nieto | Albacore tuna |
| Nishizawa [181] | North Pacific | CCA-SIED-MGET | Albatrosses |
| Oh [182] | Japan Sea | BOA | Japanese flying squid |
| Pikesley [183] | Gabon-Angola | CCA-SIED-MGET | Olive ridley turtle |
| Pikesley [184] | Cape Verde | CCA-SIED-MGET | Loggerhead turtle |
| Podesta [63] | Mid-Atlantic Bight | CCA | Swordfish |
| Reese [185] | California Current | CCA-SIED | Sardine, anchovy, herring |
| Retana [186] | San Jorge Gulf, Arg. | CCA-SIED-MGET | Marine mammals |
| Royer [187] | Gulf of Lions, Med. Sea | Canny | Bluefin tuna |
| Sabal [61] | California Current | CCA-SIED | Salmon |
| Sabarros [53] | Benguela Upwelling | CCA-SIED-Nieto | Cape gannet |
| Sagarminaga [188] | NE Atlantic | BOA | Albacore tuna |
| Santiago [67] | Indian Ocean | BOA-RG | Yellowfin tuna |
| Santiago [68] | Atlantic Ocean | BOA-RG | Yellowfin tuna |
| Sarma [48] | Arabian Sea | CCA-SIED | Zooplankton |
| Scales [152] | Celtic Sea | CCA-SIED; Miller | Gannet |
| Scales [189] | Canary Current | CCA-SIED; Miller | Loggerhead turtle |
| Scales [54] | South Atlantic | CCA-SIED; Miller | Grey-headed albatross |
| Schick [112] | Gulf of Maine | CCA | Bluefin tuna |
| Soldatini [190] | Baja California Peninsula | BOA | Black-vented shearwater |
| Sousa [191] | Gulf of Cadiz | CCA-SIED-MGET | Sunfish (mola mola) |
| Suhadha [192] | Bali Strait, Indonesia | CCA-SIED-MGET | Bali sardinella |
| Svendsen [62] | San Matias Gulf, Arg. | CCA-SIED-MGET | Various species |

**Table 1.** *Cont.*

| Reference (First Author, Ref. No.) | Region | Front Detection Algorithm/Technique | Species |
|---|---|---|---|
| Swetha [193] | Northern Indian Ocean | CCA-SIED-MGET | Potential Fishing Zone |
| Thorne [194] | NW Atlantic | CCA-SIED-MGET | Pilot whale |
| Trew [195] | Gulf of Guinea | CCA-SIED-MGET | Mammals, turtles |
| Tseng [65] | Northwest Pacific | CCA-SIED | Pacific saury |
| Varo-Cruz [196] | Eastern North Atlantic | CCA-SIED-MGET | Loggerhead turtle |
| Wall [197] | West Florida Shelf | CCA-SIED; Canny | King mackerel |
| Wang J [198] | Patagonian Shelf | Gradient | Hake |
| Wang YC [199] | South China Sea | Shimada | Ichthyoplankton |
| Wang YC [200] | East China Sea | Shimada | Ichthyoplankton |
| White [201] | Nantucket Shoals | BOA | Ducks, scoters |
| Woodson [52] | California Current | BOA | Rockfishes, invertebrates |
| Xu [143] | NE Pacific | CCA-SIED-Nieto | Albacore tuna |
| Zainuddin [202] | Makassar Strait | CCA-SIED | Skipjack tuna |
| Zhou [203] | South Pacific | BOA; CCA-SIED-Nieto | Albacore tuna |

CCA: Cayula–Cornillon algorithm [18–20]; SIED: Single-Image Edge Detection algorithm by Cayula and Cornillon [19]; MGET: Marine Geospatial Ecology Tools by Roberts et al. [136]; Miller: Multispectral composite frontal map analysis [91]; Nieto: CCA modification by Nieto et al. [92]; BOA: Belkin–O'Reilly algorithm [21]; RG: BOA R code by Galuardi (https://github.com/galuardi/boaR; accessed on 18 February 2021); Shimada: Entropy-based edge detection algorithm by Shimada et al. [90]; Canny: Edge detection algorithm by Canny [89].

Analysis of Table 1 reveals two trends. First, the entire field of front detectors is dominated by the Cayula–Cornillon algorithm (CCA), especially the Single-Image Edge Detector (SIED; [19]) implemented as an R code in the Marine Geospatial Ecology Tools (MGET) [136]. The CCA-SIED domination is especially prominent owing to the analysis by Miller [91,137] and Nieto et al.'s [92] algorithm being based on CCA-SIED.

Second, the geographical distribution of the fisheries studies included in Table 1 is strongly non-uniform. Three regions stand out: (1) the Patagonian Shelf and Shelf-Break (Argentine Sea); (2) the North Indian Ocean, especially the Arabian Sea and Bay of Bengal; (3) the Western North Pacific and its marginal seas.

The Patagonian Shelf and Shelf-Break feature several well-defined quasi-stationary year-round fronts. This region is, thus, unique in the context of the entire World Ocean, since there are no other regions with several prominent year-round fronts [2]. Fronts in the Patagonian Shelf and Shelf Break are targeted for their stocks of scallops [75,76], shrimp [166], hake [198], and various pelagic and demersal fish species [62].

The North Indian Ocean is devoid of strong quasi-stationary fronts, save for river plume fronts associated with the Ganges–Brahmaputra and Irrawaddy freshwater outflows [2,106]. Nonetheless, numerous papers by Indian, Pakistani, and Indonesian researchers are dedicated to fronts and their effects on fisheries, particularly regarding potential fishing zones (PFZ) [48,103,154,164,168,171,192,193,202], with tuna being the main target [154,168,202].

The Western North Pacific and its marginal seas feature numerous diverse fronts formed by tidal mixing, river discharge, wind-induced upwelling, and water mass convergence [2,126]. These fronts form regional ecosystems and play important roles in local fisheries [39,65,159,170,175,180,182,199,200], with major target species being tuna [180], squid [39,159,180,182], saury [65,170,180], and anchovy [175].

## 7. Discussion

**Physics:** Fronts are formed by a wide variety of physical processes. The main types of physical fronts have been known for a century, including tidal mixing fronts, water mass

convergence fronts, coastal upwelling fronts, equatorial upwelling fronts, topographic upwelling fronts, river plume fronts, and fronts of marginal ice zones [1,2]. Depending on the particular front generation mechanism, a given front can be seasonal or perennial, and their predictability in space and time varies greatly, depending on several factors. Tidal mixing fronts (TMFs) are highly predictable in space and time as tides are predictable. The locations of TMFs are largely controlled by bathymetry, which anchors such fronts. However, during each season, exact locations of TMFs depend on air–sea interactions and establishment of summertime stratification.

Some fronts change their physical nature from one season to another. For example, the Zhejiang–Fujian Front in the East China Sea [126] is a classical water mass front along the offshore boundary of the southward China Coastal Current during the wintertime northeastern monsoon season. However, during the summertime southwestern monsoon season, the China Coastal Current reverses and retreats to the north, being pushed by the southwesterly winds blowing along the Zhejiang–Fujian Coast and driving upwelling, resulting in the Zhejiang–Fujian Front becoming an upwelling front.

The persistence of fronts can be quantified by calculating the frontal frequency at a given geographical location. The analysis of frontal frequency maps in various geographical regions of the World Ocean has shown that some well-known large-scale fronts (e.g., extensions of western boundary currents) do not transpire in such maps [2,106]. This phenomenon can be explained by the increased variability of such currents and associated fronts away from the western boundaries, in the open ocean, where topographic steering is absent or extremely weak. Many persistent fronts are quasi-stationary, e.g., shelf-break fronts and tidal mixing fronts. In such cases, a long-term map of pixel-based frontal frequencies portrays such fronts as ridges of elevated frequencies. Ridge lines in such maps are assumed to be the most likely paths of the respective fronts. Examples of such robust fronts and frontal frequency maps can be found in [2,106,119–123,126].

**Biology:** Associations between fronts and biota are species-specific and critically dependent on a life stage of a given species. For example, skipjack tuna in the North Pacific spawn at the Subtropical Front (south of 30°N) but then migrate to the Subarctic Front (around 40°N) for foraging. Most fronts feature maximum biodiversity, while some fronts feature minimum biodiversity when an elevated biomass at the fronts is maintained by a few super-abundant species of fish. Each physical aspect of every front—such as its spatial and temporal scales, horizontal and vertical structures, development stage, gradients and cross-frontal ranges of physical and biochemical parameters, and spatial and temporal variability of the front—may affect different species differently and may even affect the same species differently depending on the current ontogenetic stage of the species.

**Logistics:** Comprehensive sets of frontal data (location, development stage, cross-frontal ranges of physical and biochemical parameters, vertical structure, etc.) are not readily available in the great majority of cases. Most satellites can provide either all-weather low-resolution data or cloud-contaminated high-resolution data on T, S, SSH, and CHL. Satellite-borne synthetic aperture radar (SAR) data is the only exception from the above dichotomy, as SAR provides all-weather high-resolution data. The temporal resolution of satellite data varies between 5 minutes (geostationary satellites) and 16 (Landsat)– 35 days (ERS). Data latency (time lag between data acquisition and uploading the data on the Web) is constantly improving. Currently, SST and CHL data provided by NASA and NOAA are available within 12–24 h; the agencies are aiming to reduce the data latency to 4–6 h. Information on vertical structures (stratification) can be inferred from SSH data combined with Argo buoys and oceanographic data from the World Ocean Database (https://www.nodc.noaa.gov/OC5/WOD/pr_wod.html; accessed on 18 February 2021).

**Frontal metrics and indexes:** Each snapshot of a front (obtained with in situ or satellite data in 1D, 2D, or 3D) can be characterized by several numbers such as location, vertical extent, horizontal extent, cross-frontal horizontal gradients, and ranges of physical (T, S, density) and biochemical (nutrients, oxygen concentration) variables at various depths. Fronts that separate different types of stratification need additional non-numerical descrip-

tors that reflect a qualitative change of vertical structure across the front. A comprehensive set of frontal parameters (metrics) inferred from in situ surface and subsurface data can be combined with satellite-derived frontal metrics, such as cross-frontal gradients and ranges of SST, SSS, CHL, and SSH, as well as metrics of the front's persistence, which are commonly evaluated by calculating the pixel-based frontal frequency. The most challenging problem is how to combine such multiple diverse characteristics as those enumerated above. The multispectral composite frontal map analysis technique developed by Peter Miller seems to be the most promising [91,137].

## 8. Summary

More than 80 marine ecology and fisheries studies have been identified, which have applied various front detection algorithms to satellite imagery and maps of SST, CHL, and SSH, aiming at elucidating links between fronts and biota. The vast majority of these studies confirmed previous observations of elevated biomass and biodiversity at or near fronts. In the feature-based framework, fronts are detected and characterized by a number of parameters, after which such front data are explicitly used in marine ecology and fisheries research, including statistical models of fish populations. Every front can be characterized by several parameters, including cross-frontal gradients and ranges of various quantities, both observed and derived, such as the temperature, salinity, density, dynamic height, depths of the upper mixed layer and euphotic layer, CHL concentration in the euphotic layer, and heat content. All of these front-centric parameters can be used in marine ecology and fisheries studies; however, to date, only a few such parameters have been tried and tested. A distance between an animal and the nearest front is the most popular parameter that allows quantitative studies of front–biota links. The nearest distance approach has been used in numerous studies owing to its implementation in geographic information systems. Other front parameters' roles in front–biota links are largely unexplored—the perspectives are limitless.

**Funding:** This research received no external funding.

**Data Availability Statement:** No new data were created or analyzed in this study. Data sharing is not applicable to this article.

**Acknowledgments:** The global survey of ocean fronts led by Peter Cornillon (University of Rhode Island) has been supported by the NASA (Eric Lindstrom, program manager) since 1997. The NOAA has supported the development and validation of the Belkin–O'Reilly algorithm at the URI since 2006. Cara Wilson (NOAA) promoted front applications in marine fisheries and conservation. Ken Sherman (NOAA) has included fronts as major physical components into the large marine ecosystem concept since 2005. The original manuscript was improved thanks to numerous edits, comments, and suggestions by Frederick Bingham and three anonymous reviewers. While working on this paper, the author was supported by the Zhejiang Ocean University.

**Conflicts of Interest:** The author declares no conflict of interest.

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
