# Peer review of "Remote Sensing of Ocean Fronts in Marine Ecology and Fisheries"

_remotesensing, doi:10.3390/rs13050883_

Round 1

Reviewer 1 Report

An ocean front is a relatively narrow transitional zone with enhanced horizontal gradients of physical, chemical, and biological properties that separates broader areas that differ in terms of vertical structure and stratification. In Japan, ocean fronts are generally called “shiome”, as the author mentions, which has been traditionally utilized as a proxy of fishing grounds of some pelagic species fishes. The potential relationships between ocean fronts and fishing grounds have been also taught as part of compulsory education in all Japanese elementary schools. Nevertheless, understanding of underlying mechanisms and qualitative evaluations for the front-fish relationships have not been sufficient.

Many studies that have detected ocean fronts and examined their properties have used satellite-derived data. The biggest advantage of using satellite data is that satellite sensors simultaneously measure much or even all of the horizontal structure of ocean fronts. This ability contrasts with the difficulties of making in situ measurements from ships; shipboard measurements are constrained by the number and speed of available ships. Moreover, satellite-based frontal positions can be easily overlaid onto biological distributions. Sea mammals, birds, and some pelagic fish frequently aggregate around ocean fronts.

The author summarized history of satellite-based study of ocean fronts in terms of marine ecology and fisheries, the review of which is not only comprehensive and valuable to fisheries/biological scientists but also a milestone in the continuous history.

I give the following minor comments.

(1) The author focuses on sophisticated frontal detection methods (e.g., Table 1). Characteristics for each method are briefly explained in the section 5.

My question is how users, particularly, fisheries/biological scientists should determine which method is the most appropriate. “trial and error”, “case by case”, or “depending on study purpose” may be one of the answers. However, if there is a rough guideline for the selection of these methods, it is very informative for the author to give a short note.

(2) According to a journal rule of Remote Sensing, citations in the main text are given by sequential serial number, e.g., [1].

(3) I inserted comments in the PDF manuscript and attached the file.

These are all my comments.

Reviewer 2 Report

Dear Author,

Although article topic is not my field of expertise, I want to point out that this review article was a hard work to be done - so many references cited and that's great.

Several comments regarding to the text:

- please use proper referencing in text like mentioned in RS template - square brackets with number. You used "Author(s), year" format;
- line 35 and other similar emphasizing - italics + bold + underline can't be used in combination - use only italics if you want to point out something in the text;
- line 185 and other - URL goes in chapter References, not in line with text;
- Table 1 - first column - use proper referencing format.

Best Regards.

Reviewer 3 Report

Review: Remote Sensing of Ocean Fronts in Marine Ecology and Fisheries

General Comments:

This manuscript presents a review of sensors and techniques used to detect ocean fronts from remotely-sensed data and describing how these fronts are used in fisheries research. There is a great deal of information presented and the author is an expert in this area.

The author states in the abstract and in several other places (in bold) in the paper that “A case is made for feature-based approach”. It is unclear what this approach is being contrasted with. As the author places such a strong emphasis on this approach, he should define it clearly early in the paper and tell the reader how this approach contrasts with or builds upon other approaches, which is not at all clear to me.

There is a somewhat informal tone (“these days”, “thanks to the SeaWiFS” - and others; I would avoid the use of “thanks to” in general) used in places. I think this manuscript would be stronger with a more rigorous use of language, particularly in the opening sections (1-3).

I also think this paper would benefit by more clearly defining some of the terms used throughout, such as front, eddy and upwelling early in the paper. The author should avoid the assumption that the audience is familiar with all of these terms and make sure to provide clear definitions for any terms and concepts that are described in further depth later in the paper.

The conclusion section seems not to be an actual conclusion of material presented earlier in the paper. What is presented in the conclusion here is more of a discussion and distillation of important concepts related to fronts. Perhaps this section should be labelled differently and then followed by a brief conclusion that does in fact provide a concluding summary.

I think in general that this paper would benefit from more narrative and explanation in terms of what is important and why. I think the author needs to provide more guidance for the reader by defining relevant terms (early in the paper) and pointing out what is important. What is the feature-based approach? How does it compare with other approaches? How has this approach contributed to marine fisheries research? These things are all mentioned, but I think the author should be more explicit. The author has a great deal of experience in this field of research and the list of studies is comprehensive. I recommend this for publication given some moderate revisions related to style, further definitions of key terms, and explanation/narrative on what is most important and why.

Specific Comments:

Line 9: “a feature-based approach

Line 23: replace “the fisheries” with “by fisheries researchers”

Line 28: “exploited species have been known to prefer certain temperatures” Is there a reference for this?

Line 50: “Fronts are the main structural element…”

Line 45: The author mentions and highlights in bold a “feature-based approach” here and in other places in the paper. Firstly, this should be defined. It is also unclear if there is another approach that this is in contrast to.

Line 49: A more explicit and robust definition of ocean fronts would be of benefit to the reader. Eddies and upwellings should also be defined.

Lines 61-82: It’s my view that biographical references to the author in these two paragraphs is not particularly relevant and should be removed.

Lines 91-99: The description of the paper structure might be better presented as bullet points or in outline form.

Line 113: Acronyms in parentheses should follow definitions and this should be consistent throughout the manuscript - i.e. “Coastal Zone Color Scanner (CZCS)”

Lines 101-141: The information here may be better presented as a table, but I will leave this to the authors. The lists of sensors here are also somewhat incomplete. There is no mention of newer sensors such as Sentinel sensors from the Copernicus program. SMAP should also be included in the list of SSS sensors.

Line 174: “The importance of CHL…”

Line 175: “of food webs”

Line 177: “have quickly found their usage in marine fisheries” should be replaced with “were quickly adopted for use in marine fisheries research”

Line 186: “have found its applications” should be replaced with “have been applied”

Line 189: It’s not entirely accurate to state that “SST and CHL characterize just a very thin surface layer”. For SST this is true, but CHL does contain some depth-integrated information to the range of 20-30m in clear water. The author should make this distinction.

Line 246: “Such stratification boundaries profoundly affect sound propagation at sea, thereby shaping ocean soundscapes”. This needs to referenced.

Line 576: “Main types of physical fronts have been …”
